# LiDAR-to-Radar Translation Based on Voxel Feature Extraction Module for Radar Data Augmentation

**DOI:** 10.3390/s24020559

**Published:** 2024-01-16

**Authors:** Jinho Lee, Geonkyu Bang, Takaya Shimizu, Masato Iehara, Shunsuke Kamijo

**Affiliations:** 1Emerging Design and Informatics Course, Graduate School of Interdisciplinary Information Studies, The University of Tokyo, 4 Chome-6-1 Komaba, Meguro City, Tokyo 153-0041, Japan; bang@kmj.iis.u-tokyo.ac.jp; 2Mitsubishi Heavy Industries Machinery Systems Ltd., 1-1, Wadasaki-cho 1-chome, Hyogo-ku, Kobe 652-8585, Japan; takaya.shimizu.hz@mhi.com; 3Mitsubishi Heavy Industries Ltd., 1-1, Wadasaki-cho 1-chome, Hyogo-ku, Kobe 652-8585, Japan; masato.iehara.ht@mhi.com

**Keywords:** LiDAR-to-Radar translation, voxel feature extraction, radar data augmentation, autonomous driving

## Abstract

In autonomous vehicles, the LiDAR and radar sensors are indispensable components for measuring distances to objects. While deep-learning-based algorithms for LiDAR sensors have been extensively proposed, the same cannot be said for radar sensors. LiDAR and radar share the commonality of measuring distances, but they are used in different environments. LiDAR tends to produce less noisy data and provides precise distance measurements, but it is highly affected by environmental factors like rain and fog. In contrast, radar is less impacted by environmental conditions but tends to generate noisier data. To reduce noise in radar data and enhance radar data augmentation, we propose a LiDAR-to-Radar translation method with a voxel feature extraction module, leveraging the fact that both sensors acquire data in a point-based manner. Because of the translation of high-quality LiDAR data into radar data, this becomes achievable. We demonstrate the superiority of our proposed method by acquiring and using data from both LiDAR and radar sensors in the same environment for validation.

## 1. Introduction

In autonomous vehicles, one of the most crucial functionalities is environmental perception. Measuring distance is the key function for environmental perception. The primary sensors for distance measurement in autonomous vehicles are LiDAR and radar. LiDAR measures distance based on light, while radar uses radio waves for distance measurement. Due to these characteristics, LiDAR is heavily affected by environmental elements, leading to significant distance measurement errors during conditions like rain or fog. Radar, on the other hand, is less affected by environmental factors, but its radio wave nature tends to generate noisy data due to reflections from various obstacles, posing challenges for distance measurement. For these reasons, obtaining quantified data from LiDAR and radar is challenging, especially for radar. The scarcity of radar data has even impacted the development of state-of-the-art deep learning technologies, resulting in a significa nt gap in the number of algorithms using radar data compared to those using image data.

Deep learning technology is one of the most prominent algorithms receiving attention in recent years. Thanks to an unprecedented amount of data and computational resources, it can outperform traditional algorithms by learning from vast amounts of data in parallel. Deep learning plays a crucial role in various tasks, including object detection [1,2,3], classification [4,5,6], and segmentation [7,8,9]. The performance of deep learning algorithms is directly proportional to the amount of data that are available. Therefore, when vast data sets like images and LiDAR are publicly available, it becomes possible to develop high-performance algorithms. On the contrary, radar data have limited availability in public data sets. Thus, constraints are imposed on radar data-driven algorithm development.

As mentioned earlier, the issue of data scarcity significantly impacted the development of deep learning, leading to the recent proliferation of research on data augmentation. The foundation for research in data augmentation was laid by Generative Adversarial Networks (GANs) [10]. GANs [10] have made significant contributions, not only to data augmentation but also to image translation, resulting in a variety of derivative research proposals [11,12,13,14,15] in this field. Some studies [16,17,18] have proposed GAN-based methods for LiDAR data augmentation. However, these studies have primarily remained limited to image and LiDAR data and have not extended to augmenting radar data.

To address these challenges, we propose a deep-learning-based LiDAR-to-Radar translation network, which consists of voxel feature extraction modules that enable the translation from LiDAR to radar data. LiDAR and radar sensors organize data in the form of point clouds and are mainly used to measure distance. The point clouds from each sensor represent a collection of points that indicate the locations where light or radio waves are reflected from objects. The term “translation” is commonly used in the field of image translation, often referring to research where images are transformed into images of different styles, such as day2night and sunny2rain image translation. In this study, we define “translation” as the transformation of LiDAR points into radar points. LiDAR and radar are both in point format, and this can be considered a kind of style transformation. Figure 1 illustrates the purpose of this study. The voxel feature extraction module is designed based on voxel feature encoding (VFE) [19], which divides the point cloud into 3D voxels of uniform intervals and transforms the point groups within each voxel into integrated feature representations. In essence, our proposed method compresses LiDAR point data into a format that resembles radar point data. This is possible because both LiDAR and radar sensors acquire point data through reflections from objects.

We validated our approach using data collected at JARI, which include simultaneous LiDAR and radar data acquired in the same environment. JARI stands for the Japan Automobile Research Institute. It provides controllable experimental environments, including factors like rainfall and fog density, to acquire the necessary data for autonomous driving. The JARI data set consists of relatively simple driving scenarios. To demonstrate the effectiveness of the proposed method in more complex driving scenarios, we acquired and validated data from real-world scenarios in Tokyo (i.e., a Tokyo data set). Our method significantly enhances radar data augmentation by generating radar data from a relatively large volume of LiDAR data. Furthermore, since LiDAR data are nearly noise-free, we observed that the radar data transformed from LiDAR exhibit significantly less noise compared to actual radar data. We believe that our proposed method can make significant contributions not only to radar data augmentation but also to a driving simulator, which can verify various autonomous driving algorithms before being applyied to those in the real world.

The contributions of the proposed method are as follows:We introduce the first, innovative, deep-learning-based LiDAR-to-Radar translation method that preserves 3D point information.We make a significant contribution to radar data augmentation, which, in turn, greatly advances the development of deep-learning-based algorithms that utilize radar data.The proposed method utilizes LiDAR and radar data acquired in the same environment for training and quantitative evaluation. The experimental results demonstrate that it is possible to successfully generate radar data with less noise from reliable LiDAR data, compared to actual radar data.

The remainder of this paper is structured as follows: In Section 2, we discuss related work, including general image translation, LiDAR/radar translation, and LiDAR-to-radar translation. Section 3 outlines our proposed methodology, while Section 4 presents the experimental results. We conclude and discuss future research directions in Section 5.

## 2. Related Work

In this section, we will introduce various related translation researches, categorizing them into three main areas: general image translation, LiDAR and radar translation within same sensor domain, and LiDAR-to-radar translation for radar data augmentation. For general image translation, we note the original image generation model and explain various image generation models derived from that model. In LiDAR and radar translation within same sensor domain, we will delve into research concerning data augmentation for LiDAR and radar, which stems from image translation methods. Here, we will introduce studies focused on transformations within the same type of sensor, including reproducing changes in LiDAR data due to climatic effects and transforming simulated data into real LiDAR data. Lastly, in LiDAR-to-Radar translation for radar data augmentation, we introduce research on transformations from LiDAR to radar data. As our proposed method focuses on the conversion from 3D LiDAR data to 3D radar data, we present research that is closely related to this research.

### 2.1. General Image Translation

The inaugural neural network for data generation, introduced by Goodfellow et al., was the Generative Adversarial Network (GAN) [10]. Various GAN-based image translation methods have emerged from the foundation of the GAN [10], including Pix2Pix [14]. This is a representative supervised image translation method that uses conditional GANs (cGANs) [11] to map inputs to output images. Similar methods, with supervision, are used for various tasks like generating realistic outdoor scenes from semantic labels [20], and converting sketches to photographs [21]. Although these supervised image translation methods excel in terms of accuracy, they require paired data in the source and target domain. Due to this, they struggle when no supervised data are available for generated images. In unsupervised image translation, which does not require paired data, two prominent methods are CycleGAN [15] and DualGAN [22]. They address domain translation by introducing cycle consistency. CoGAN [23] considers a shared latent space for advanced image translation. Several studies focus on multi-domain image translation [24,25], achieving impressive results across various tasks. In this study, we use the paired data of LiDAR and radar for training, which falls under the category of supervised learning.

### 2.2. LiDAR and Radar Translation within Same Sensor Domain

While previous research has predominantly revolved around image data, there are also numerous deep-learning-based translation methods focusing on data augmentation using LiDAR. An exemplary study is [16], which delves into LiDAR translation. However, it has a noteworthy limitation: the loss of height information due to the representation of 3D point cloud data as 2D Bird-Eye-View (BEV) data. The BEV generated using LiDAR data typically represents a projection of a three-dimensional environment, which is changed into a two-dimensional perspective viewed from above. Recent investigations have begun to address this limitation by considering weather changes and height information into LiDAR translation, as evidenced in [17,18]. Despite image and LiDAR data dominating the translation research landscape, radar data have received comparatively less attention. Only a handful of research studies are dedicated to radar data augmentation. One prominent example is [26], which harnesses CNNs in tandem with domain-specific data-augmentation techniques. However, this approach is time-consuming and best-suited to a limited scope of targets. Another radar data augmentation technique, introduced by Sheeny, M. in 2020, is parameterized radar data augmentation (RADIO) technique [27]. While it holds significance in the realm of radar augmentation employing deep learning methods, it confines its testing to simple target objects within small data sets and may lack stability, especially in comprehensive road environments. The proposed study centers its attention on radar augmentation and makes a substantial contribution by generating radar data derived from high-quality LiDAR data, thereby elevating the field of radar data augmentation.

### 2.3. LiDAR-to-Radar Translation for Radar Data Augmentation

The research field of LiDAR-to-Radar translation is relatively less explored compared to other translation studies. One of the most notable studies in this area is L2R GAN [28]. L2R GAN [28] generates radar spectra for natural scenes using a provided LiDAR point cloud. This translates to radar spectra images based on the provided LiDAR Bird’s Eye View (BEV). Here, radar spectrum image refers to an image that visually represents the frequency spectrum of radar data, serving as a form of visualization and analysis for radar data. The model acquires radar-data-generation skills through the use of an occupancy grid mask as a guiding element. Additionally, it incorporates a suite of local region generators and discriminator networks into its architecture. This unique combination empowers L2R GAN [28] to leverage the strengths of global features while retaining intricate details at the local region level. As a result, it excels not only in capturing the large-scale cross-modal relationships between LiDAR and radar but also in refining fine-grained details.

Figure 2 illustrates the inputs and outputs of L2R [28] and the proposed method. L2R, as previously explained, takes 3D LiDAR point clouds and transforms them into Bird’s Eye View (BEV) 2D images, excluding height information, for input. Originally, BEV 2D images are widely used for visualizing 3D point clouds. The format of its output is radar spectrum image, which excludes height information from 3D radar point clouds. In other words, L2R’s training involves comparing 2D BEV images from LiDAR with 2D radar spectrum images for learning. The advantage of the above process is that the data used for training are 2D data similar to images, making it convenient to apply existing GAN-based image translation models. However, it has a drawback in that it loses height information from the 3D point cloud and lacks reliability, since it converts LiDAR to radar data without height information. Therefore, the proposed method, as shown in Figure 2, is designed for LiDAR-to-Radar translation based on voxel feature extraction, preserving 3D point information. With the proposed method, it becomes possible to maintain 3D point information, resulting in more reliable radar data generation from LiDAR and reducing radar data noise.

## 3. Proposed Method

In this section, we introduce the proposed method. The key component of our method is the voxel feature extraction module. Our ultimate goal is to generate realistic radar data from reliable LiDAR data. Initially, we designed our approach to be based on sensor-independent data, so we provide an explanation regarding the training data format. Considering the distinct LiDAR and radar data formats is an essential step, as conventional translation methods were originally designed for image translation purposes. In our proposed method, LiDAR uses x, y, z coordinates, and intensity values, while radar uses x, y, z coordinates and RCS (Radar Cross Section) values. Detailed differences between these formats will be described later. Additionally, the range of values in LiDAR and radar data is entirely different from the RGB values of an image. To address these challenges, we designed a network that includes the voxel feature extraction module.

### 3.1. Voxel Feature Extraction Module

To extract feature information from the 3D point cloud, we first divide the point cloud into 3D voxels with the same spacing. Then, we use the voxel feature extraction module to create features for each voxel based on the points contained within the voxel. The first piece of research related to the voxel feature extraction module was Voxel Feature Encoding (VFE) of VoxelNet [19] in 2017. Still, various studies [29,30,31] continue to propose novel approaches based on VoxelNet. The original VoxelNet [19] integrates the features of voxels by 3D convolution [32,33] to create local voxel features. These features are then input into the Region Proposal Network (RPN) [34,35] to generate bounding boxes for 3D object detection. However, in this study, we only employ VFE to extract information from the LiDAR point cloud at the voxel level. Therefore, we do not utilize 3D convolution and RPN.

We show a simple outline of our voxel feature extraction module based on Voxel Feature Encoding (VFE) in Figure 3. The conventional VFE approach forwards complete features to fully connected layers and subsequently applies element-wise max-pooling across the features of all points within the voxel to extract voxel-wise features. In other words, the features extracted after pooling represent the values associated with individual voxels. However, inputting all the features simultaneously into the fully connected layer can result in the loss of positional information. Hence, we address this issue by segregating coordinate features (x, y, z) and point value features (intensity values for LiDAR and RCS values for radar) in our proposed voxel feature extraction module. LiDAR intensity, collected for each point, quantifies the strength of the laser pulse’s return signal, influenced by the object’s reflectivity. Radar cross-section (RCS) is a metric that quantifies a target’s capacity to reflect radar signals toward the radar receiver. It represents the ratio of backscatter power per steradian (i.e., a unit of solid angle) in the direction of the radar to the power density intercepted by the target. Through our proposed voxel feature extraction module, LiDAR’s coordinate values become comparable to radar’s coordinate values, and LiDAR’s intensity values are gradually adjusted to align with radar’s RCS values. We will provide more detailed information about the proposed voxel feature extraction module later, when explaining the pipeline.

### 3.2. Pipeline

Here, we will delve into how our proposed method converts LiDAR data into radar data. We illustrate a simplified pipeline in Figure 4. The proposed approach comprises five main components: Voxel Partition, Grouping, Random Sampling, Stacked Voxel Feature Extraction, and Voxel Feature Sampling.

#### 3.2.1. Voxel Partition

As mentioned earlier, the first step is to divide the given point cloud into equally sized cuboids called voxels. This process is known as voxelization. Denote the z-axis length of the area where LiDAR points are distributed as *D*, the y-axis length as *H*, and the x-axis length as *W*. Also, note that the dimensions of the voxel in the z, y, and x directions are defined as $v_{D}$, $v_{H}$, and $v_{W}$, respectively. Then, the entire point cloud will be divided into a 3D voxel grid with the following dimensions:(1)$$\begin{array}{c} D^{\prime }=D/v_{D}, \end{array}$$
(2)$$\begin{array}{c} H^{\prime }=H/v_{H} \end{array}$$
(3)$$\begin{array}{c} W^{\prime }=W/v_{W}, \end{array}$$

In this paper, the settings were chosen as *D* = 8, *H* = 40, *W* = 200, $v_{D}$ = 1.0, $v_{H}$ = 2.0, $v_{W}$ = 2.0 for stable training and performance. However, these settings may need to be adjusted depending on the data set that is used.

#### 3.2.2. Grouping

This process involves assigning points to their respective voxels. Consequently, each point corresponds to a specific voxel, and different voxels may contain varying numbers of points due to the characteristics of LiDAR data.

#### 3.2.3. Random Sampling

Processing a LiDAR point cloud with around 70,000 points directly requires tremendous computational resources. This can result in bias due to the significant variation in point density, depending on the location. Therefore, we introduce a maximum point count per voxel, denoted as *T*. For voxels with more than *T* points, only *T* points are sampled and retained. In this paper, we set *T* to 45. This helps reduce computational demands, mitigate point density imbalances, and introduce variation during training.

#### 3.2.4. Stacked Voxel Feature Extraction

This part is the core network part of the proposed method, as shown in Figure 3. First, the input LiDAR data are in the form of (x, y, z, r), where (x, y, z) represents 3D coordinates, and ‘r’ represents the reflectance (i.e., intensity of the LiDAR beam reflection). To process these data, we calculate the centroid of LiDAR points within each voxel as the average of their 3D coordinates. Next, we compute the relative positions of each point with respect to this centroid and concatenate this information with the original data.

The proposed stacked voxel feature extraction builds upon the VFE approach [19]. In contrast to the conventional VFE, we split this into separate Fully Convolution Networks (FCNs) [36] for coordinate and point value instead of using a simple FCN. We feed concatenated features of per-point characteristics and relative coordinate characteristics within the voxel into these separate FCNs. As a result, the features of all points within the voxel are transformed to obtain point-wise features. Both the coordinate and point value FCNs consist of a linear layer, batch normalization, and leaky ReLU. We apply average-pooling to the coordinate-specific point-wise feature to obtain information that is close to the center of the voxel, and we use max-pooling for the point-specific point-wise feature to extract the strongest point value representing the voxel. When max-pooling was applied to the coordinate-specific point-wise feature instead of average-pooling, the coordinate error increased by 9.224 cm compared to the baseline. Conversely, when average-pooling was applied to the point-specific point-wise feature instead of max-pooling, the RCS error increased by 14.091. Based on these results, it was confirmed that applying average-pooling to coordinate-specific and max-pooling to point-specific point-wise features is optimal. The features that underwent pooling are concatenated to output the final voxel-wise feature.

The proposed voxel feature extraction module includes two sets of a linear layer, batch normalization, and leaky ReLU. Through experimental results, it has been confirmed that the optimal number of modules in this study is two. In other words, there are a total of four sets of a linear layer, batch normalization, and leaky ReLU. Each FCN has channel numbers of 64, 128, 64, and 32, and the coordinate features retain one-fourth of the total features.

#### 3.2.5. Voxel Feature Sampling

The output features from the stacked voxel feature extraction module are transformed to resemble radar values, but there are too many candidate values. The output of the stacked voxel feature extraction module typically ranges from approximately 900 to 1200, whereas actual radar data consist of only 80–250 values, with many potentially representing noise. To address this issue, it is necessary to sample the extracted voxel features. In voxel sampling, we first select features that fall within the measurement range of the radar. Then, the final features where the coordinate and point value features are concatenated are sampled as the top *K* based on the point value features. In this study, *K* was set to 130 for the JARI data set and 200 for the Tokyo data set, based on a statistical analysis of the number of points in the radar data set.

### 3.3. Training

The training process in this study is similar to traditional supervised learning-based translation training. Firstly, during loss calculation, both real radar values and radar values generated by the proposed method are used. Of course, the generated radar values are derived from real LiDAR values. In conventional translation methods, typically used for image-to-image translation, features like RGB values per pixel are used for loss calculation. Common loss functions in this context include Mean Squared Error (MSE) and Mean Absolute Error (MAE). Image-based translation methods can use such loss calculations because the input and output sizes are either the same or can be adjusted. However, in the proposed method, the number of input and output points is never guaranteed to be the same. While the proposed point padding method discussed later can alleviate this issue to some extent, there is no guarantee that the input and output point arrays will be of the same size. To address this problem, the proposed research uses KL-Divergence [37,38] as the loss function. KL-divergence loss measures how different two result distributions are. A lower value indicates that the two distributions are more similar. KL-Divergence is calculated as follows:(4)$$\begin{array}{c} KL\left(P||Q\right)=\sum_{c=1}^{M}P_{c}\log\frac{P_{c}}{Q_{c}} \end{array}$$

Note that *P* indicates real radar data and *Q* means-generated radar data from LiDAR data by the proposed method. *M* is the number of all radar points.

In conventional image translation, real and generated images typically have identical image sizes. Even if they are not exactly the same, they are resized to match is a straightforward process. This is crucial because it allows for loss calculations when the sizes are uniform. However, when dealing with radar point data, resizing is not a viable option, especially when the number of real and generated radar points differs. To address this issue, we propose a method called “point padding” to equalize the number of sampled generated features and real radar data, facilitating the loss calculations mentioned earlier.

Conventional image padding methods [39] typically fill the edges with a minimum RGB value of 0. Drawing inspiration from this, in our point-padding approach, we set the minimum RCS value (i.e., −65) at the maximum and minimum x and y coordinates, representing the edge coordinates of the radar points. This ensures that the padded points have a minimal impact on the learning process. This methodology not only enables loss calculations but also contributes to stable and effective learning. We utilized the Leaky Rectified Linear Unit (Leaky ReLU) [40] instead of the Rectified Linear Unit (ReLU) [41] as the activation function, because RCS values can be negative and we aim to represent a wider range of point values.

## 4. Experimental Results

To conduct the following experiments, it is essential to determine which values to use from the LiDAR and radar data. Our proposed method aims to be sensor-agnostic, utilizing only data values that can be acquired by any LiDAR or radar sensor. From LiDAR data, we extract the x, y, z coordinates, and intensity values. The intensity data in LiDAR, collected for each point, serve as a measurement of the laser pulse’s return signal strength, which is influenced by the object’s reflectivity with which the laser pulse interacts. For radar data, we utilize the x, y, z coordinates, along with the radar cross-section (RCS) values. The radar cross-section (RCS) is a measurement that quantifies a target’s ability to reflect radar signals back towards the radar-receiver. In other words, RCS is defined as the ratio of backscatter power per steradian (a unit of solid angle) in the direction of the radar to the power density intercepted by the target. Radar’s main point values include doppler information in addition to RCS. However, since there is no corresponding information in LiDAR that can be directly matched with radar’s Doppler information, we excluded Doppler information during LiDAR-to-Radar translation. We transformed the values from both LiDAR and radar into an array of shape (N, 4) for use in our training. Here, N represents the number of points in both LiDAR and radar, and 4 corresponds to the x, y, z coordinates, and point values (i.e., intensity for LiDAR and RCS for radar).

We conducted three experiments in total. The first experiment aimed to determine the optimal network architecture by comparing performance based on the number of voxel feature extraction modules. The second experiment involved a qualitative evaluation, where we visually compared actual radar data with the radar data generated from LiDAR to facilitate a more in-depth discussion. In the final experiment, we conducted an objective quantitative evaluation by comparing the positions and RCS values between actual radar data and radar data generated from LiDAR. Furthermore, we conducted a quantitative performance comparison based on whether the coordinate values were separated or not.

### 4.1. Data Set

We used the JARI data set as our data set, which was employed in our previous research on LiDAR translation [17,18] that considered environmental factors. The JARI data set is a reliable data set, to the extent that it was submitted to this journal as part of the LiDAR translation [17] research last year. In our previous data set, we acquired the only LiDAR data by adjusting the levels of rain and fog in the experimental space provided by Japan Automobile Research Institute (JARI). However, for this study, since environmental factors like rain and fog are not relevant, we collected data differently. The JARI data set includes relatively straightforward driving scenarios. To showcase the efficacy of our proposed method in more intricate driving situations, we also collected and validated data from real-world scenarios in Tokyo (i.e., Tokyo data set).

We attached LiDAR and radar sensors to the vehicle, as shown in Figure 5. LiDAR sensors were mounted on the top of the vehicle, while radar sensors were positioned at the front license plate location. The LiDAR sensor used for the proposed method is the Ouster LiDAR with 64 vertical channels. A meticulous calibration process was executed to ensure precise data collection. We acquired data in clear weather conditions, ensuring the absence of environmental factors. The environments where the data were collected presented numerous obstacles, including actual vehicles and road markings (e.g., white lines). Figure 6 shows an example of the JARI and Tokyo data set. In Figure 6, the images, LiDAR data, and radar data are displayed from left to right. The upper figures display data for the JARI data set, and the below figures show those for the Tokyo data set. The LiDAR and radar data were visualized using the Rviz tool. Radar images indicate radar points as green, red, and blue dots with LiDAR data visualization. Note that the radar data contain numerous noise data points, which appear regardless of the object’s location. This is not a sensor defect issue but rather an inherent characteristic of radar data due to their properties.

### 4.2. Experiment 1: The Most Optimal Architecture

While there are numerous image translation networks available for reference, there is no universally recognized architecture for LiDAR-to-Radar translation. LiDAR-to-Radar translation that particularly preserves the 3D information of points represents a novel and pioneering research field. As a result, we conducted an experiment to determine the most appropriate architecture for LiDAR-to-Radar translation.

The core architecture of LiDAR-to-Radar translation proposed in this research is the voxel feature extraction module. The voxel feature extraction module is based on VFE, as explained earlier. However, the proposed module is configured to separate coordinate values (i.e., x, y, z coordinates) and point values (i.e., intensity for LiDAR and RCS for radar) for more accurate position learning. The number of voxel feature extraction modules, which is constructed with (Multilayer Perceptron (MLP) [42,43], has a significant impact on the performance of LiDAR-to-Radar translation. Also, since the MLP is composed of multiple Fully Convolution Networks (FCNs), the module can be viewed as a set of FCNs. When too few modules are used, the learning may be incomplete. However, an overfitting [44,45] problem can occur when too many modules are used. Therefore, in the first experiment, we compared the performance depending on the number of voxel feature extraction modules to establish the optimal architecture.

We provide a comparison of the results based on the number of voxel feature extraction modules used. The training data set utilized for this comparison contains the Sunny–5 km/h data, which correspond to the data collected under clear weather conditions with no environmental factors while the vehicle maintained a constant speed of 5 km/h, as depicted in Figure 6. This data set encompasses 471 scenes. Additionally, the test data set, also based on the Sunny–40 km/h data, comprises 122 scenes with a constant speed of 40 km/h. We confirmed that the most effective architecture for LiDAR-to-Radar translation in the JARI data set was determined to be two voxel feature extraction modules by comparing RCS error results.

The results were obtained by calculating the errors to average the sum of the differences between the ground truths (i.e., real radar data) and the generated results (i.e., radar data generated from real LiDAR data). The equation is given as follows:(5)$$\begin{array}{c} Error_{v}=\sum_{1}^{M}\left\{\sum_{1}^{N}\left|Real_{v_{N}}-Fake_{v_{N}}\right|\right\} \end{array}$$

The error value is denoted as $Error_{v}$. This is categorized by different types of value, represented by *v*, which include x, y, z coordinates, and RCS. Here, *M* represents the total number of scenes, and *N* represents the number of points in each scene. Furthermore, $Real_{v_{N}}$ and $Fake_{v_{N}}$, respectively, denote the ground truth (i.e., real radar data) and the generated results (i.e., radar data generated from actual LiDAR data). The RCS error results for the voxel feature extraction modules when there are 1, 2, and 3 modules were 24.28, 10.35, and 54.89, respectively. In other words, the architecture with two voxel feature extraction modules is the most effective. We found that when there is only one voxel feature extraction module, there are insufficient weights for the training of over 60,000 LiDAR data points. Additionally, when there are three or more voxel feature extraction modules, overfitting occurs, causing the generated RCS values to converge to a constant value.

The primary role of the voxel feature extraction module is the encoding of LiDAR points. There are various methods for encoding LiDAR points, aside from the voxel feature extraction module. To assess the superiority of the voxel feature extraction module, we compared it with different representative feature extraction methods, such as original fully connected layer-based Multilayer Perceptron (MLP) [42,43] and 1 × 1 convolution layers [46], for coordinate and RCS error comparison, as shown in Table 1. The comparative experiment was conducted using the JARI data set. To ensure an objective comparison, the sizes of filters, kernels, and other parameters for each method were standardized. As shown in Figure 7, a qualitative evaluation was also conducted.

From the results in Table 1 and Figure 7, it is evident that the proposed method outperforms other encoding techniques. In the case of the original MLP, a high coordinate error was observed, which is reflected in the qualitative evaluation, demonstrating a bias in the generated points. This indicates that an MLP-based encoding approach alone is not suitable for effectively learning location information. In terms of 1 × 1 convolution, it performs better in terms of location information compared to MLP, but generates a significant amount of noise data, resulting in a high RCS error value. Furthermore, both original MLP and 1 × 1 convolution struggle to reproduce radar points at the vehicle’s location. In contrast, the proposed method excels in replicating radar points at the vehicle’s location and produces less noisy data. This is attributed to the voxel-based approach, which allows for superior location information reproduction, and the separation of coordinate features, making it more effective in learning location information.

### 4.3. Experiment 2: Qualitative Evaluation of LiDAR-to-Radar Translation

Using the optimized architecture, it is important to verify whether radar data are generated naturally, with low noise, from reliable real LiDAR data. In the second experiment, we qualitatively compared the generated radar data with actual radar data acquired in the same environment for verification. Figure 8 presents a comparison with the JARI data set, between actual radar data and radar data generated from LiDAR. In Figure 8, from left to right, the display includes a real image, a real LiDAR, a real radar, and a fake radar created by our proposed method. The height values of both the real and generated radar data are predominantly close to 0. Consequently, both the real and generated radar data are visualized as seen from above, and projected onto the z-axis at z = 0. The number on the results indicates the distance to the vehicle in front. In the radar data, the green squares represent radar points corresponding to the location of the vehicle. The color indicates RCS values, with blue representing lower RCS values and red indicating higher RCS values. In radar results, the coordinate in the center-left represents the position of the radar sensor. The distance increases, from the left to the right. In the real radar data of Figure 8, the point cluster that spreads horizontally represents the walls of the experimental space. However, in the generated radar data, the points representing walls are not reproduced as accurately. This discrepancy is because data are sampled based on the top *K* RCS values in the voxel feature sampling process. If we set a larger number for *K*, it would be possible to replicate more closely, but this would also include points with lower reliability.

The JARI data set played a crucial role in shaping the optimized architecture. However, since the JARI data set consists of relatively simple scenarios and data within constrained spaces, relying solely on JARI data to demonstrate the superiority of the proposed method would be insufficient. Therefore, we conducted additional qualitative evaluations using the Tokyo data set. The Tokyo data set comprises more complex scenarios and real-world data compared to the JARI data, allowing for us to establish the versatility and excellence of the proposed method. Figure 9 displays a comparison using the Tokyo data set, between actual radar data and radar data generated from LiDAR. From left to right, the visualizations include a real image, a real LiDAR, a real radar, and a radar synthetized by our proposed method. The height values in both the actual and generated radar data are largely concentrated around 0. As a result, both the real and synthesized radar data are represented as if observed from above, with a projection onto the z-axis at z = 0. We show data from three scenarios in the Tokyo data set. In the radar data, the green squares represent radar points corresponding to the location of vehicles and bus. Also, fences at construction sites with high reflectivity are marked as purple areas. The color scale represents RCS values, where blue corresponds to lower RCS values and red represents higher RCS values. In the radar visualizations, the center-left coordinate denotes the radar sensor’s position.

RCS values in actual radar data typically fall within the range from −65 to 65. However, it is important to note that these RCS values are logarithmic in nature. Thus, negative RCS values can be considered as very low values close to zero and are interpreted as noise. Radar sensors inherently contain reliability challenges. Due to the scattering of radio waves, radar sensors may generate low RCS values even in the absence of reflective objects. As shown in Figure 8 and Figure 9, the blue points represent lower RCS values. Real data often include numerous low RCS points. In contrast, our proposed method leverages high-quality LiDAR data for reference and generates radar data with higher RCS values, thereby ensuring reliability. As shown in Figure 8 and Figure 9, the green rectangles in the radar data indicate the radar points corresponding to the position of vehicles and the bus positions. Also, fences at construction sites with high reflectivity are marked as purple areas in Figure 9. The radar data generated by the proposed method better represent the radar points of vehicles, bus, and fences compared to the actual radar. This is also thanks to the high-reliability LiDAR points. Our approach benefits from voxel feature sampling that takes into account the radar sensor’s measurement range. The voxel feature sampling prevents the generation of noise data outside the radar’s range. This adds to the proposed method’s robustness and reliability.

### 4.4. Experiment 3: Quantitative Evaluation of LiDAR-to-Radar Translation

For an objective evaluation, we conducted quantitative assessments using the optimal architecture with two voxel feature extraction modules, as mentioned earlier. The data sets used include the JARI and Tokyo data sets. For the JARI data set, we utilized 471 scenes from the Sunny–5 km/h data, which were collected under clear weather conditions with no environmental factors while the vehicle maintained a constant speed of 5 km/h, as our training data set. For testing, we employed the Sunny—40 km/h data set, comprising 122 scenes with a constant speed of 40 km/h. In an environment where all conditions are adjustable, Tokyo data acquisition is not feasible, like in the JARI data set. Therefore, we used data from 700 scenes for both training and testing, all obtained in clear weather conditions. Table 2 and Table 3 show the quantitative comparison results of the JARI and Tokyo data sets, respectively. The results were obtained using the same methodology as described in Equation (Equation 5) of Experiment 1. This involved calculating errors by averaging the sum of the differences between the ground truths, which represent real radar data, and the generated results, which are radar data generated from actual LiDAR data, for x, y, z, and RCS values. As with actual radar data, the z coordinates of the generated radar data are also close to 0. To compare the performance between the traditional voxel feature extraction module and the proposed one, we present the results of both modules. The traditional voxel feature extraction module combines coordinate features and point value features, while our proposed method separates them. Thus, we refer to these as the “Not-segregated Module” and the “Segregated Module,” respectively, as shown in Table 2 and Table 3.

As shown in Table 2 and Table 3, the traditional method exhibits significant errors in coordinate accuracy, whereas our proposed method achieves x, y, and z coordinate errors within 10 cm. This result demonstrates that the proposed LiDAR-to-Radar translation method effectively captures and reflects location information with high accuracy. The actual value and error of z are nearly zero; however, there is an over 4.7% difference in the overall coordinate error between training with and without z-values. This signifies that height values also significantly impact coordinate learning. Furthermore, we confirmed the natural conversion of LiDAR intensity values to radar RCS values through our proposed method, as evidenced by the RCS error values. Compared to the traditional method, the improved performance in terms of RCS values suggests that separating and learning only the point value features in our proposed approach contribute to its enhanced performance.

For the additional quantitative evaluation, we compare the number of noise radar points and radar points within the range of objects, such as vehicles and buses, between actual radar data and generated radar data. As previously explained, real data include a significant amount of noise, whereas our approach, which generates radar data from reliable LiDAR data, results in significantly less noise. Furthermore, the number of radar points within the range of objects like vehicles and buses is higher in the proposed method compared to actual data. These results are presented in Table 4. The quantitative evaluation was conducted on both the JARI and Tokyo data sets. Table 4 provides the average number of total radar points, noise points, and points within the range of objects, per one scene on each data set. Noise points were counted for RCS values between −65 and −55, and the considered objects include high-reflectivity vehicles, and buses.

Considering the comprehensive evaluation of the results from Experiments 1, 2, and 3, it is evident that our proposed LiDAR-to-Radar translation method successfully converts LiDAR data into a format that aligns with the characteristics of radar data.

### 4.5. Discussion

Our proposed LiDAR-to-Radar translation method, as demonstrated through the experiments we presented earlier, was validated to generate natural radar data with relatively fewer noise artifacts compared to real radar data. This was achieved by utilizing high-quality LiDAR data as a reference, resulting in a more reliable and realistic radar data output. Furthermore, we confirmed the effectiveness of the voxel feature extraction module proposed in our approach for LiDAR-to-Radar translation. This module plays a crucial role in achieving successful translation from LiDAR to radar data, as demonstrated in our experiments. We have high expectations that our proposed method will make a substantial contribution to radar data augmentation. Additionally, paired data sets of LiDAR and radar data can be provided for various applications through the proposed method. These contributions are expected to address the limitations posed by the scarcity of radar data in the development of autonomous driving algorithms.

However, it is important to acknowledge that our proposed research is not without its limitations. One notable aspect is that the initial weights at the start of training tend to have a significant impact. This means that if the initial weights differ significantly from the ideal weights, the training process can be time-consuming. In some cases, it may even be more efficient to restart the training process from scratch. Our future goal is to make the voxel size adaptable based on the distribution of LiDAR points. We expect that this flexibility will significantly enhance the proposed method’s versatility and applicability, making it more adaptable to a wide range of scenarios and data sets.

## 5. Conclusions

Our study introduces a novel approach for performing LiDAR-to-Radar translation to enhance radar data augmentation. Traditional methods for LiDAR-to-Radar translation primarily focus on translation based on the Bird’s Eye View (BEV) representation. However, these methods cannot effectively incorporate height information from LiDAR due to the inherent characteristics of BEV. In contrast, our proposed method excels in handling radar translation by leveraging high-quality LiDAR data while preserving height information. In other words, the proposed method is the first deep-learning-based LiDAR-to-Radar translation method maintaining 3D information.

The key to achieving this is our innovative voxel feature extraction module, which is built upon the foundation of traditional Voxel Feature Encoding (VFE). Since VFE primarily relies on fully connected layers, accurately forwarding positional features to deep layers poses a challenge. To tackle this problem, our proposed voxel feature extraction module segregates positional features from entire features, enhancing the model’s ability to learn intricate positional information. In addition to this module, we devised several strategies (e.g., point padding for loss calculation) to ensure stable training, ultimately enabling us to perform reliable LiDAR-to-Radar translation.

In our experiments, we utilized a data set that we collected ourselves. These data were acquired within the same environment, enabling our proposed method to effectively perform supervised translation. We observed that our proposed model was capable of generating natural radar data with significantly fewer noisy artifacts compared to real radar data.

It is worth noting that our study extends beyond mere radar data augmentation. LiDAR and radar sensors are both essential for distance measurement in autonomous driving, and share similar objectives. However, there is a notable shortage of paired open data sets for these sensors. Paired data sets are invaluable resources for autonomous driving research and can play a pivotal role in advancing the development of fully autonomous driving systems. Therefore, our research not only addresses the issue of data scarcity but also provides a valuable tool for fellow researchers in the field. Furthermore, we believe that our proposed method can make significant contributions to a driving simulator which can verify various autonomous driving algorithms before their application in the real world.

In our future endeavors, we aim to enhance the proposed method, adapting it to a wider range of environments, such as various open data sets (e.g., nuScenes [47]). Given that we have, thus far, tested the proposed method with a limited data set, our plans also involve collecting diverse data from actual roads. Subsequently, we intend to refine and optimize the proposed method to better suit the challenges posed by various real road conditions. Also, we plan to develop the ability to adjust the voxel size based on the distribution of LiDAR points. We anticipate that this increased flexibility will greatly improve the versatility and applicability of the proposed method, making it suitable for a broad spectrum of scenarios and data sets. Finally, we plan to devise an advanced LiDAR-to-Radar translation method using KP-Convolution [48], which is a encoding module that allows for convolution to be applied directly to the point cloud without transformation. KP-convolution [48] appears to be a promising candidate for enhancing the performance of the proposed method, even though it may require a significant amount of time for training.

## Figures and Tables

**Figure 1 sensors-24-00559-f001:**
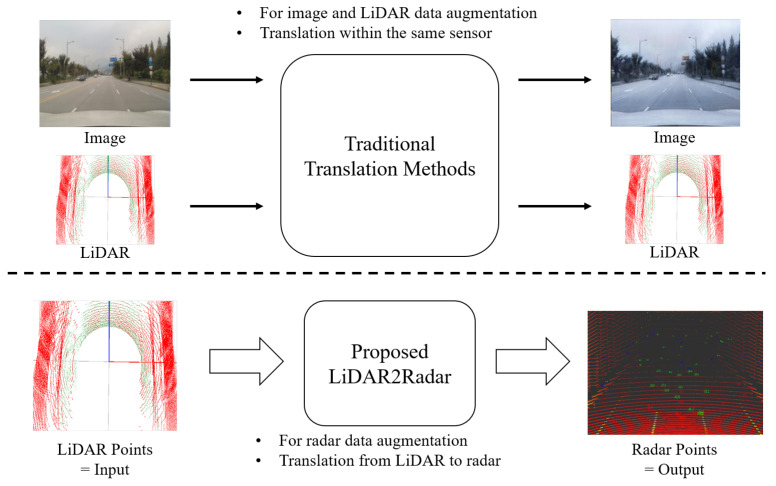
The purpose of this study. We propose a network based on a voxel feature extraction module that enables the translation from LiDAR to radar data. The term “translation” is frequently employed within the realm of image translation, typically denoting studies that convert images into various styles. Nonetheless, in this research, our interpretation of “translation” encompasses the conversion of LiDAR points into radar points. Given that LiDAR and radar data both employ a shared point format, this could be regarded as a sort of style transformation. While many traditional translation methods are limited to image and LiDAR data, our proposed method is specifically designed for augmenting radar data. Furthermore, most traditional methods focus on translation within the same sensor domain, whereas our research enables translation between different sensor domains (i.e., LiDAR-to-Radar).

**Figure 2 sensors-24-00559-f002:**
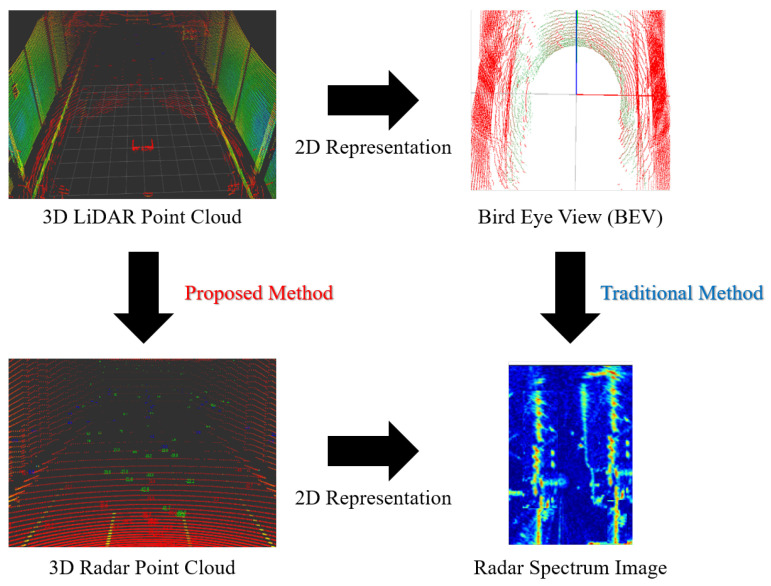
Main difference between the proposed method and L2R [28] in the data format. L2R takes 3D LiDAR point clouds and converts them into 2D Bird’s Eye View (BEV) images, excluding height information. The output format of L2R is a radar spectrum image, which omits the height information from 3D radar point clouds. L2R’s training process involves comparing 2D BEV images from LiDAR with 2D radar spectrum images to facilitate learning. However, a drawback of this approach is the loss of height information from the 3D point cloud. To address this limitation, we developed a new approach for LiDAR-to-Radar translation that is based on voxel feature extraction, preserving the 3D point information. This novel method allows for us to retain the 3D point information, resulting in the generation of more reliable radar data from LiDAR and a reduction in radar data noise.

**Figure 3 sensors-24-00559-f003:**
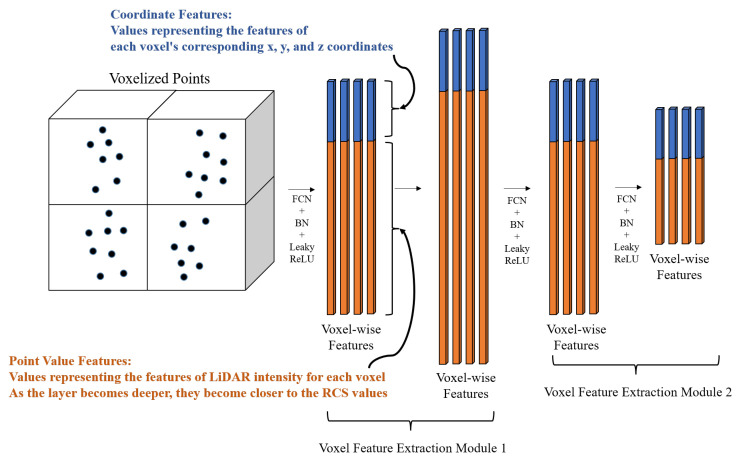
Outline of our voxel feature extraction modules based on VFE. The traditional VFE forwards entire features to fully convolution networks (FCNs). The dark blue dots indicate the LiDAR points. Then, it performs element-wise max-pooling on the features of all points within the voxel to obtain voxel-wise features. However, inputting all the features simultaneously into the fully connected layer can lead to the loss of positional information. Therefore, we split these into separate FCNs for coordinate and point value, instead of using a simple FCN. The coordinate features highlighted in blue are values representing the features of each voxel’s corresponding x, y, and z coordinates. Also, the point value features highlighted in orange are values representing the features of LiDAR intensity for each voxel. As the layer becomes deeper, they become closer to the RCS values. Average-pooling and max-pooling are applied to each of the FCNs to efficiently extract the features of the coordinates and point values, respectively. The proposed voxel feature extraction module includes two sets of linear layers, batch normalization, and leaky ReLU activation functions. Through experimentation, it has been confirmed that the optimal number of such modules is 2. Therefore, there are a total of four sets of linear layers, batch normalization, and leaky ReLU activation functions. The number of channels in each FCN is 64, 128, 64, and 32, with the coordinate features retaining a quarter of the total features.

**Figure 4 sensors-24-00559-f004:**
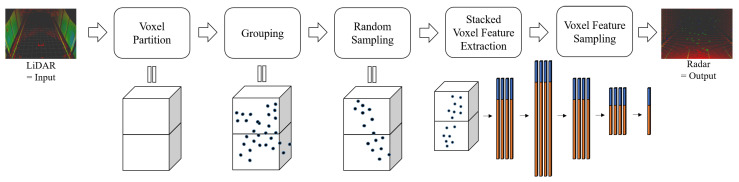
A brief pipeline of the proposed LiDAR-to-Radar translation. The proposed method can be divided into five major elements: Voxel Partition, Grouping, Random Sampling, Stacked Voxel Feature Extraction, and Voxel Feature Sampling. Through these processes, it is possible to generate radar data with relatively low noise from reliable LiDAR data. This method not only significantly contributes to radar data augmentation but also plays a crucial role in creating a reliable LiDAR-radar paired data set. The dark blue dots indicate the LiDAR points. Blue and orange bars represent FCN for coordinate features and point value features, respectively.

**Figure 5 sensors-24-00559-f005:**
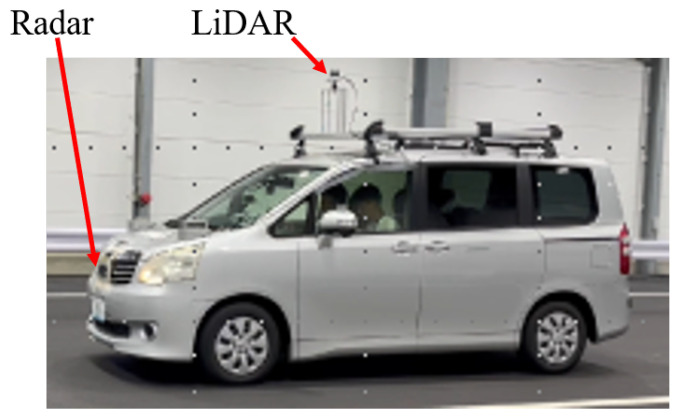
Sensor configuration of the experimental vehicle for data acquisition of the JARI and Tokyo data set. LiDAR and radar sensors were attached to the top of the vehicle and in the front license plate position, respectively. The LiDAR sensor employed in the proposed method is the Ouster LiDAR with 64 vertical channels. Precise calibration was carried out to ensure accurate data collection.

**Figure 6 sensors-24-00559-f006:**
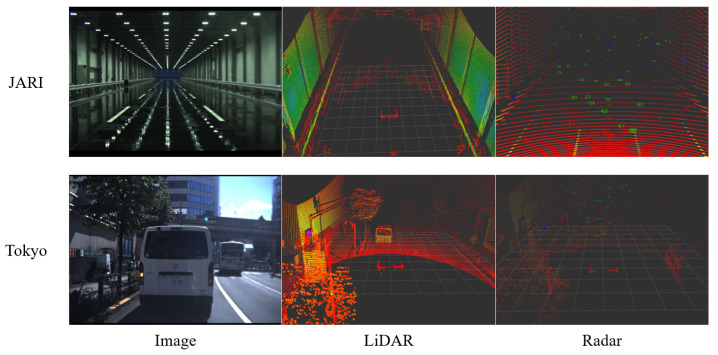
An example of the JARI and Tokyo data set. The upper and below figures present the image, LiDAR, and radar data from left to right, for the JARI and Tokyo data set, respectively. The LiDAR and radar data are displayed as visualized in the Rviz tool. Radar images indicate radar points as green, red, and blue dots with LiDAR data visualization. It is important to note that the radar data contain numerous noise data points, which appear regardless of the object’s location. This issue is not attributed to a sensor defect but rather stems from inherent characteristics of radar data.

**Figure 7 sensors-24-00559-f007:**
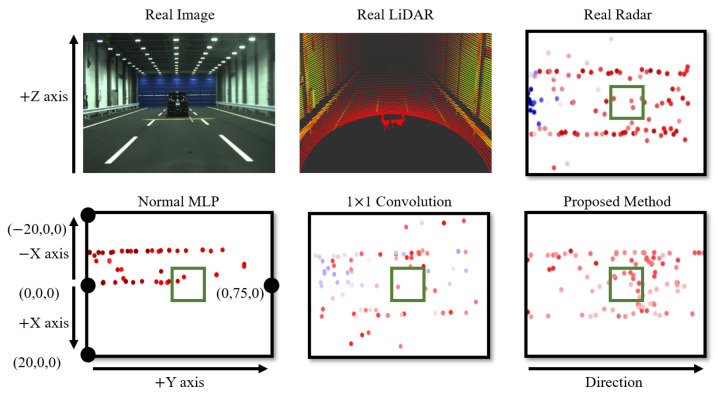
Qualitative comparison results depending on the encoding methods. From the top left to right, it shows a real image, a real LiDAR, a real radar. From bottom left to right, it demonstrates a fake radar generated by original MLP, 1 × 1 convolution, and the proposed method, respectively. The height values of both the real and generated radar data are predominantly close to 0. Consequently, both the real and generated radar data are visualized as seen from above, projected onto the z-axis at z = 0. Green squares in the radar data indicate radar points at which the vehicle is located. The color-coding represents RCS values, where blue corresponds to lower RCS values, and red signifies higher RCS values. Within the radar results, the coordinate shown in the center-left indicates the position of the radar sensor. Moving from left to right, the distance from the sensor increases.

**Figure 8 sensors-24-00559-f008:**
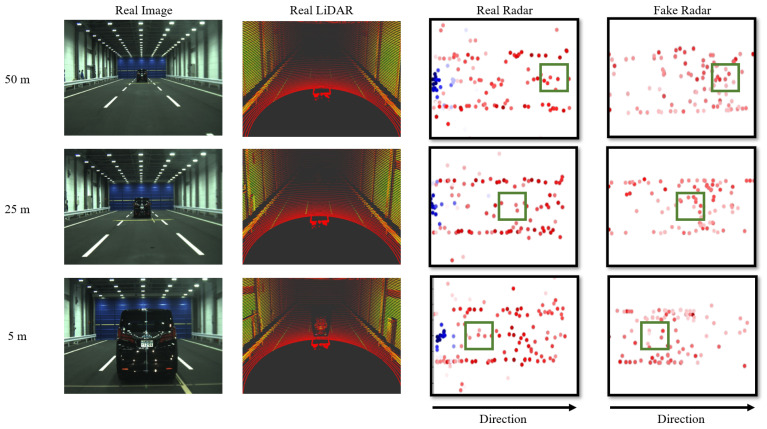
Examples of comparison results using the JARI data set between actual radar data and radar data generated from LiDAR. From left to right: a real image, a real LiDAR, a real radar, and a fake radar generated by the proposed method. The height values of both the real and generated radar data are predominantly close to 0. Consequently, both the real and generated radar data are visualized as seen from above, projected onto the z-axis at z = 0. The number on the results denotes the distance to the vehicle ahead. Green squares in the radar data indicate radar points where the vehicle is located. The color coding represents RCS values, where blue corresponds to lower RCS values, and red signifies higher RCS values. Within the radar results, the coordinate shown in the center-left indicates the position of the radar sensor. Moving from left to right, the distance from the sensor increases.

**Figure 9 sensors-24-00559-f009:**
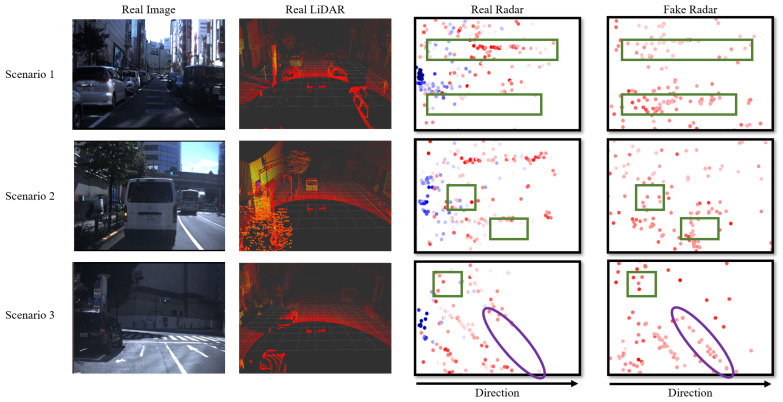
Examples of comparison results using Tokyo data set, between actual radar data and radar data generated from LiDAR. From left to right: a real image, a real LiDAR, a real radar, and a fake radar generated by the proposed method. The height values in both the actual and generated radar data are largely concentrated around 0. As a result, both the real and synthesized radar data are represented as if observed from above, with a projection onto the z-axis at z = 0. We demonstrate data from three scenarios in the Tokyo data set. Green squares in the radar data indicate radar points where the vehicles and bus are located. The purple areas in the radar data represent fences at construction sites with high reflectivity. The color coding represents RCS values, where blue corresponds to lower RCS values, and red signifies higher RCS values. Within radar results, the coordinate shown in the center-left indicates the position of the radar sensor. Moving from left to right, the distance from the sensor increases.

**Table 1 sensors-24-00559-t001:** Quantitative comparison results depending on the different feature extraction modules. The coordinate and RCS errors indicate the error values, derived from the average of summing the errors between the ground truths (i.e., real radar data) and the generated results (i.e., radar data generated from real LiDAR data).

Feature Extraction Methods	Original MLP	1 × 1 Convolution	Proposed Method
Coordinate Errors	28.711	11.492	7.931
RCS Errors	12.775	20.389	10.356

**Table 2 sensors-24-00559-t002:** Comparison results using the JARI data set, between the ground truth, which represents real radar data, and the generated results, which are radar data generated from actual LiDAR data, for x, y, z coordinates and RCS error values. The each error value is obtained by calculating errors as the average sum of the differences between the ground truths and the generated results. To compare the performance between the traditional voxel feature extraction module and the proposed one, we present the results of both modules. The traditional voxel feature extraction module combines coordinate features and point value features, while our proposed method separates them. Therefore, we refer to them as the “Not-segregated Module” and the “Segregated Module,” respectively.

Voxel Feature Extraction Module	X Coordinate Error [cm]	Y Coordinate Error [cm]	Z Coordinate Error [cm]	RCS Error
Not-Segregated Module	10.759	25.512	0.016	12.217
Segregated Module	3.044	7.324	0.021	10.356

**Table 3 sensors-24-00559-t003:** Comparison results using the Tokyo data set, between the ground truth, which represents real radar data, and the generated results, which are radar data generated from actual LiDAR data, for x, y, z coordinates and RCS error values. Each error value is obtained by calculating errors by averaging the sum of the differences between the ground truths and the generated results. To compare the performance between the traditional voxel feature extraction module and the proposed one, we present the results of both modules. The traditional voxel feature extraction module combines coordinate features and point value features, while our proposed method separates them. Therefore, we refer to them as the “Not-segregated Module” and the “Segregated Module,” respectively.

Voxel Feature Extraction Module	X Coordinate Error [cm]	Y Coordinate Error [cm]	Z Coordinate Error [cm]	RCS Error
Not-segregated Module	13.116	24.735	0.031	17.947
Segregated Module	5.948	6.881	0.017	9.744

**Table 4 sensors-24-00559-t004:** Quantitative evaluation to compare the number of noise radar points and radar points within the range of objects, such as vehicles and buses, between actual radar data and generated radar data. Quantitative evaluation was carried out on both the JARI and Tokyo data sets. This Table presents the average number of total radar points, noise points, and points falling within the range of objects, per one scene on each data set. Noise points were counted for RCS values between −65 and −55, and the considered objects include high-reflectivity vehicles, and buses.

	Total Radar Points	Noise Radar Points	Object Radar Points
JARI (Real)	137	39	4
JARI (Generated)	135	2	9
Tokyo (Real)	201	54	31
Tokyo (Generated)	200	3	47

## Data Availability

The data used in this study are subject to restrictions and are available upon request. Interested parties can obtain the data from the corresponding authors by making a request. Please note that the data cannot be publicly accessed due to contractual agreements related to the project.

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
