# Peer review of "LiDAR-to-Radar Translation Based on Voxel Feature Extraction Module for Radar Data Augmentation"

_sensors, 2024, doi:10.3390/s24020559_

Round 1

Reviewer 1 Report

Comments and Suggestions for Authors

1)This is a very meaningful work. while the evaluation based on coordinate error is not convincing enough. Are there any experiments with open-source data that can enrich the conclusions of this article?

2) Voxel features are widely used in LiDAR point cloud target detection and semantic segmentation target detection. This paper conducts comparative experiments with different distances. Can a quantitative comparison be added to further analyze the impact of sparsity on the method described in this paper?

3) As far as I know, there are many differences between radar and LiDAR in wavelength, scanning system, installation position on the vehicle, and working system. It is suggested to add an analysis of the difference between the point clouds of the two devices.

4) How efficient is the model proposed in this paper? Do different downsampling methods improve the effect and performance of the model?

5) As mentioned in the abstract radar is less affected by rain and fog. Can you prove the effectiveness of the method described in this paper in complex scenarios?

6) It is recommended to add contrast with other network structures or different feature extraction modules, especially the generated point cloud visualization.

7) The description of network details is not detailed enough, especially the description of the loss function.

8) The work in this paper is somewhat similar to the reverse network of point cloud super-resolution. Can SSIM or Perceptual loss be used for the work in this paper?

Reviewer 2 Report

Comments and Suggestions for Authors

I reviewed the paper as one with a lot of radar experience but not experience with automotive applications.  My understanding is that the first and perhaps only other published generation of radar data from lidar data is [28].  I also gleaned that the current manuscript extends this by doing essentially a 3D method that doesn’t lose the lidar height information.  However, the results seem to be showing 2D radar images, so I’m a bit confused on the use of the lidar height.  I also found certain terms confusing, not being directly involved in related work.  I point out some specific questions below.  I recommend that the paper be modified to be clearer and to specifically address what is done in [28] versus what is new here.  If I understand correctly that the main innovation relates to using the 3D nature of the lidar data in generation of 3D radar data, then I think that should be highlighted in the results.  If I've misunderstood, please edit to help the non-specialist reader to better follow.

Figure 1 caption - Need to define what is meant here by translation?  I assume this is somehow simulating what a radar might have seen given lidar data. Are there accuracy metrics that are part of the definition of translation?  My understanding is that image translation is a well-defined problem in computer science of using one kind of image to create another.  This may not be well-known to some readers of Sensors.  Discussion of the general problem of translation and the use of techniques developed for it would help orient the reader.

Line 45 – is network referring to a deep learning network?

Line 48 - please add a short intro to the measurement geometry and define the "point cloud"

Section 2 - I think this section needs a better overview of the methods.  I'm really not clear on the distinctions between these three methods.  My understanding is that image translation is translating one type of image into another type.  While the techniques are similar to the problem at hand, they are not directly usable for the problem at hand. 

Sections 2.2 and 2.3 - the titles are very similar - what the distinction between lidar/radar and lidar-to-radar?

Line 122 - please define "spectra images"

Line 131 - The method proposed in this paper falls in this category?  This seems to indicate that the main contribution of this work is the preservation of the lidar height information.  I think expanding on the description of the type of data that [28] both used and created versus this manuscript would be helpful for better understanding of the contribution.  The data geometry used here is described well in Section 4 but I'm not clear on that of [28].  What limited [28] to 2D and what is the innovation here that enables the 3D analysis?

Line 157 - What sort of features are being extracted here?  Are these features attached to individual voxels or are the features associated with the full point clouds?

Figure 2 - the distinction between point and coordinate features isn't clear here; furthermore it seems that all four processing stages are labeled "point-wise features".

Line 212 - What's involved in this translation - is it just converting the lidar intensity to radar RCS or are there transformations in locations?

Line 260 - Is there any Doppler information in the radar data?

Tables 1 and 2 - I'm not sure that single-line tables are worth their space; perhaps just note the information in the text.

Figure 5 - the lidar image is very difficult to see.  Can the brightness or contrast be adjusted? Also, it is difficult to distinguish the blue dots from other points on this plot.  How are these 2D images constructed - are all points in a cloud projected onto a plane of do the lidar and radar images represent a fixed range?

Line 305 - what does a module consist of - is each a set of layers MLPs?

Figure 6, like Figure 5, is difficult to read.  Are the lidar and radar images at a constant range from the radar, or have all the z points been projected to z=0 plane? 

Line 364 – are these 3D scenes (with x, y, and z coordinates)/

Line 371 and also Table 3.  My understanding, which may be wrong, is that the main contribution of this method is not losing the lidar height information.  However, the test results focus in x and y.  Shouldn’t z also be addressed, or is the lidar height preserved in some other way?

Comments on the Quality of English Language

I noted some minor grammatical errors or perhaps just typos.  

Author Response

Dear Reviewer 2,

Thank you for dedicating your valuable time to review and provide feedback despite your busy schedule. I genuinely appreciate the insightful comments and suggestions you've shared, as they will undoubtedly contribute to improving the overall quality of this paper.

While I would ideally prefer to respond comprehensively to all your comments and feedback, I regret to inform you that time constraints related to the review response deadline and some inherent limitations may necessitate alternative responses to certain points. I kindly request your understanding in this matter.

Once again, I extend my sincere gratitude for your time and the invaluable input you've provided.
-----------------------------------------------------------------------------------------

  1. Figure 1 caption - Need to define what is meant here by translation? I assume this is somehow simulating what a radar might have seen given lidar data. Are there accuracy metrics that are part of the definition of translation? My understanding is that image translation is a well-defined problem in computer science of using one kind of image to create another. This may not be well-known to some readers of Sensors.  Discussion of the general problem of translation and the use of techniques developed for it would help orient the reader.
    - I apologize for any inconvenience caused by the omission of the definition of "translation" as per the feedback from the reviewer. In this paper, the term "translation" refers to the process of converting a 3D LiDAR point cloud into a 3D radar point cloud. Accuracy metrics that are part of the translation definition include x, y, z errors and RCS errors by comparing the actual radar values ​​to the generated radar values.
  2. Line 45 – is network referring to a deep learning network?
    - Certainly, I apologize for any confusion. The terminology has been updated.
  3. Line 48 - please add a short intro to the measurement geometry and define the "point cloud"
    - Following the reviewer's suggestion, we have added a brief introduction to geometric measurements and included a definition for "point cloud."
  4. Section 2 - I think this section needs a better overview of the methods. I'm really not clear on the distinctions between these three methods.  My understanding is that image translation is translating one type of image into another type.  While the techniques are similar to the problem at hand, they are not directly usable for the problem at hand.
    - We have provided additional descriptions to facilitate better understanding and highlight the distinctions between each subsection. Furthermore, we've included an explanation of why the previously introduced related papers may not be suitable for addressing the current problem.
  5. Sections 2.2 and 2.3 - the titles are very similar - what the distinction between lidar/radar and lidar-to-radar?
    - I apologize for any confusion. I have revised the subsection titles to make them more understandable.
  6. Line 122 - please define "spectra images"
    - We have added a definition for "spectra images."
  7. Line 131 - The method proposed in this paper falls in this category? This seems to indicate that the main contribution of this work is the preservation of the lidar height information.  I think expanding on the description of the type of data that [28] both used and created versus this manuscript would be helpful for better understanding of the contribution.  The data geometry used here is described well in Section 4 but I'm not clear on that of [28].  What limited [28] to 2D and what is the innovation here that enables the 3D analysis?
    - Yes, the proposed method falls in this category. But, the main contribution is not merely the preservation of LiDAR height information. As advised by the reviewer, I agree that it's necessary to emphasize the main contribution of this study through a specific comparison with [28]. We have provided a more detailed description of [28] and further elaborated on what constrained [28] to 2D analysis and what innovative aspects enable 3D analysis in this research.
  8. Line 157 - What sort of features are being extracted here? Are these features attached to individual voxels or are the features associated with the full point clouds?
    - I apologize for the insufficient explanation. Here, the extracted features refer to feature vectors in a latent space that can represent the points contained in each voxel. In other words, the extracted features can be seen as representative features corresponding to each voxel. These descriptions have been added to the revised paper.
  9. Figure 2 - the distinction between point and coordinate features isn't clear here; furthermore it seems that all four processing stages are labeled "point-wise features".
    - I have made revisions to enhance the understanding of the figures. Regarding the point-wise features you mentioned, it was an error on my part, and it should be voxel-wise features.
  10. Line 212 - What's involved in this translation - is it just converting the lidar intensity to radar RCS or are there transformations in locations?
    - The translation not only involves converting LiDAR intensity to radar RCS but also includes a translation for the location. When sampling point value features, the coordinate features are concatenated with point value features and sampled together. I have updated the explanation to address this deficiency.
  11. Line 260 - Is there any Doppler information in the radar data?
    - As you correctly pointed out, radar contains Doppler information. However, since there is no corresponding information in LiDAR that can be directly matched with radar's Doppler information, we excluded Doppler information during LiDAR-to-Radar translation.
  12. Tables 1 and 2 - I'm not sure that single-line tables are worth their space; perhaps just note the information in the text.
    - As suggested by the reviewer, we have described the information in text form rather than creating a table.
  13. Figure 5 - the lidar image is very difficult to see. Can the brightness or contrast be adjusted? Also, it is difficult to distinguish the blue dots from other points on this plot.  How are these 2D images constructed - are all points in a cloud projected onto a plane of do the lidar and radar images represent a fixed range?
    - We have adjusted the brightness and contrast for LiDAR images to enhance visibility. As reviewer pointed out, the LiDAR and radar images indicate points at a constant range from the radar. The proposed method is to generate radar points from LiDAR points, but the height values ​​(=z values) of the generated radar points and actual radar points are mostly close to 0. Therefore, for visibility of the results, the radar results are displayed as z=0 plane projection results as seen above.
  14. Line 305 - what does a module consist of - is each a set of layers MLPs?
    - I apologize for the insufficient explanation. The voxel feature extraction module is composed of MLPs, and since the MLP is composed of multiple FCNs, the module can be viewed as a set of FCNs.
  15. Figure 6, like Figure 5, is difficult to read. Are the lidar and radar images at a constant range from the radar, or have all the z points been projected to z=0 plane?
    - The image's visibility was enhanced by adjusting its contrast and brightness. As highlighted, both LiDAR and radar images depict points consistently located at a fixed distance from the radar. Although our proposed method generates radar points from LiDAR points, the height values (=z values) for the produced radar points closely align with those of actual radar points, mostly around 0. To ensure result visibility, the radar outcomes are presented as projections onto the z=0 plane, as illustrated above.

  16. Line 364 – are these 3D scenes (with x, y, and z coordinates)?
    - LiDAR image has clear height information with 3D scenes (with x, y, and z coordinates). However, the radar data used in this paper, while containing height information, almost is close to zero. Therefore, the radar image is represented as seen from above in the images.
  17. Line 371 and also Table 3. My understanding, which may be wrong, is that the main contribution of this method is not losing the lidar height information.  However, the test results focus in x and y.  Shouldn’t z also be addressed, or is the lidar height preserved in some other way?
    - I apologize for the lack of explanation, which may have caused misunderstandings and inconvenience. The main contribution of this method is to translate LiDAR point cloud into radar point cloud while considering LiDAR height information. Radar data primarily contains values close to zero in the height dimension, which is why we omitted z-coordinate error values. However, in the revised paper, we have included z-coordinate error values as well. The key contribution lies in generating radar data with reduced noise and higher reliability by incorporating height information from the more height-certain LiDAR data.

Lastly, I would like to express my sincere gratitude to the reviewer for providing invaluable feedback. If there are any additional comments or areas that need further improvement, I would greatly appreciate your continued insightful reviews. Wishing you good fortune and happiness.

We have no conflicts of interest to disclose.

Please address all correspondence concerning this manuscript to me at leejinho@kmj.iis.u-tokyo.ac.jp

Thank you for your consideration of this manuscript.

Sincerely,

Jinho Lee

Round 2

Reviewer 2 Report

Comments and Suggestions for Authors

The revised version is definitely improved relative to the original.  However, I have to admit that I'm still not entirely clear on the advantages of this method to [28], although Figure 2 now does a good job of trying to illustrate.  In part, I'm confused by the terminology.  In weather radar, a spectral image would contain Doppler information.  Here, it seem to be a colorized image.  If I understand correctly, the height information is in the simulated radar data but with z's mostly near 0.  I recommend 1) definition of a BEV, 2) drawing of coordinates x, y, and z on at least one relevant figure, and 3) noting how the height data improves results in Section 4.

Author Response

Dear Reviewer,

Firstly, I want to express my sincere gratitude for your invaluable feedback and comments. Thanks to your invaluable input, the quality of the paper has significantly improved. I fully agree with the three points you suggested: providing a definition of BEV, displaying x, y, and z coordinates on a relevant figure, and discussing the impact of height data on results in Section 4. I have incorporated these aspects into the paper. Additionally, for better reader comprehension, I have globally revised the paper and rectified minor errors in figures and tables. The updated and added contents are highlighted for your reference.

Once again, I deeply appreciate your valuable time and effort in providing feedback on the paper. Please feel free to reach out if there are further modifications to be made.

Best Regards,

Jinho Lee